# GRADIENT BASED MEMORY EDITING FOR TASK-FREE CONTINUAL LEARNING

## ABSTRACT

Continual learning often assumes a knowledge of (strict) task boundaries and identities for the instances in a data stream—*i.e.*, a "task-aware" setting. However, in practice it is rarely the case that practitioners can expose task information to the model; thus needing "task-free" continual learning methods. Recent attempts towards task-free continual learning focus on developing memory-construction and replay strategies such that model performance over previously seen instances is best retained. In this paper, looking from a complementary angle, we propose to "edit" memory examples to allow for the model to better retain past performance when memory is replayed. Such memory editing is achieved by making gradient updates to memory examples so that they are more likely to be "forgotten" by the model when viewing new instances in the future. Experiments on five benchmark datasets show our proposed method can be seamlessly combined with baselines to significantly improve performance and achieve state-of-the-art results. [1]

## 1 INTRODUCTION

Accumulating past knowledge and adapting to evolving environments are one of the key traits in human intelligence (McClelland et al., 1995). While contemporary deep neural networks have achieved impressive results in a range of machine learning tasks Goodfellow et al. (2015), they haven't yet manifested the ability of continually learning over evolving data streams (Ratcliff, 1990). These models suffer from catastrophic forgetting (McCloskey & Cohen, 1989; Robins, 1995) when trained in an online fashion—*i.e.*, performance drops over previously seen examples during the sequential learning process. To this end, continual learning (CL) methods are developed to alleviate catastrophic forgetting issue when models are trained on non-stationary data streams (Goodfellow et al., 2013).

Most existing work on continual learning assume that, when models are trained on a stream of tasks sequentially, the task specifications such as task boundaries or identities are exposed to the models. These task-aware CL methods make explicit use of task specifications to avoid catastrophic forgetting issue, including consolidating important parameters on previous tasks (Kirkpatrick et al., 2017; Zenke et al., 2017; Nguyen et al., 2018), distilling knowledge from previous tasks (Li & Hoiem, 2017; Rannen et al., 2017), or separating task-specific model parameters (Rusu et al., 2016; Serrà et al., 2018). However, in practice, it is more likely that the data instances comes in a sequential, non-stationary fashion without task identity or boundary—a setting that is commonly termed as task-free continual learning (Aljundi et al., 2018). To tackle this setting, recent attempts on task-free CL methods have been made (Aljundi et al., 2018; Zeno et al., 2018; Lee et al., 2020). These efforts revolve around regularization and model expansion based approaches, which rely on inferring task boundaries or identities (Aljundi et al., 2018; Lee et al., 2020) and perform online paramater importance estimation (Zeno et al., 2018), to consolidate or separate model parameters.

In another line of efforts, memory-based CL methods have achieved strong results in task-free setting Aljundi et al. (2019b). These methods store a small set of previously seen instances in a fix-sized memory, and utilize them for replay (Robins, 1995; Rolnick et al., 2019) or regularization (Lopez-Paz & Ranzato, 2017; Chaudhry et al., 2019a). The core problem in memory-based CL methods is how to *manage* the memory instances (*e.g.*, which to replace with new instances) and *replay* them given a restricted computation budget, so that the model performance can be maximally preserved or

---

[1]Code has been uploaded in the supplementary materials and will be published.

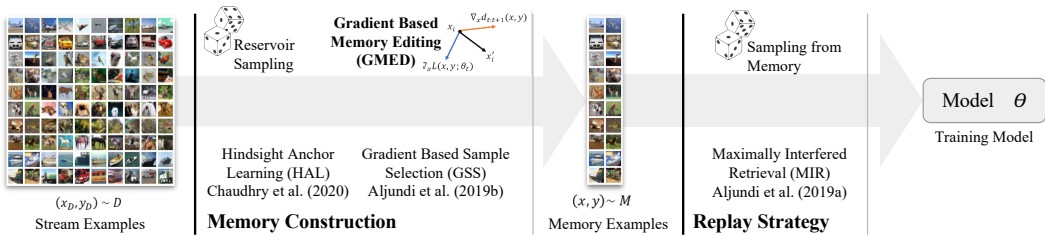

Figure 1: Categorization of memory-based methods on Task-free Continual Learning. Reservoir sampling and Sampling from Memory are the ways that Experience Replay (ER) uses to construct and replay memory respectively. In the recent line of works, Gradient based Sample Selection (GSS) and Hindsight Anchor Learning (HAL) explored ways to construct memory, while Maximally Interfering Retrieval (MIR) focused on replay strategy. Our method, Gradient-based Memory Editing (GMED) falls under the former category but provides a new angle to memory construction.

enhanced. Prior work developing these methods have tried to either identify: 1) what instances to include in memory from a data stream (Aljundi et al., 2019b; Rebuffi et al., 2017; Chaudhry et al., 2019b); and 2) which instances in memory need to be replayed at what training step (Aljundi et al., 2019a).

In this paper, we provide a new approach to solving the memory management problem in task-free continual learning by studying how to make gradient updates on stored memory examples. We develop a novel memory editing algorithm which complements existing memory-replay methods and data-sampling strategies for memory management (updates). The challenge is to propose a plausible and sound optimization objective of editing. We employ the same intuition as previous study (Toneva et al., 2019; Chaudhry et al., 2020; Aljundi et al., 2019a): examples that are likely to be forgotten should be prioritized. Our proposed method, named Gradient-based Memory EDiting (**GMED**), edits examples stored in the memory with gradient-based updates so that they are more likely to be forgotten. Specifically, we estimate the "forgetting" of a stored example by its loss increase in the upcoming one online model update. Finally, we perform gradient ascent on stored examples so that they are more likely to be forgotten.

Experiments show that our algorithm consistently outperforms baselines on five benchmark datasets under various memory sizes. Our ablation study shows the proposed editing mechanism outperforms alternative editing strategies such as random editing. We demonstrate that the proposed algorithm is general enough to be used with other strong (more recent) memory-based CL methods to further enhance performance, thus allowing for improvements in many benchmark datasets.

## 2 RELATED WORKS

**Task-aware Continual Learning.** Most of continual learning algorithms are studied under "task-aware" settings, where the model visits a sequence of clearly separated "tasks". A great portion of algorithms make explicit use of task boundaries (Kirkpatrick et al., 2017; Rusu et al., 2016; Lopez-Paz & Ranzato, 2017), by learning separate parameters for each task, or discourage changes of parameters that are important to old tasks. Existing continual learning algorithms can be summarized into three categories: regularization-based, architecture-based and data-based approaches. Regularization based approaches (Kirkpatrick et al., 2017; Zenke et al., 2017; Nguyen et al., 2018; Adel et al., 2020) discourage the change of parameters that are important to previous data. Model expansion-based approaches (Rusu et al., 2016; Serrà et al., 2018; Li et al., 2019) allows expansion of model architecture to separate parameters for previous and current data. Data-based approaches (Robins, 1995; Shin et al., 2017; Lopez-Paz & Ranzato, 2017) replay or constrain model updates with real or synthetic examples.

**Task-free Continual Learning.** Recently, task-free continual learning (Aljundi et al., 2018) have drawn increasing interest, where we do not assume knowledge about task boundaries. To the best of our knowledge, only a handful number of regularization based (Zeno et al., 2018; Aljundi et al., 2018), model-expansion based (Lee et al., 2020), generative replay based (Rao et al., 2019), continual meta-learning and meta-continual learning (He et al., 2019; Caccia et al., 2020; Harrison et al., 2020) approaches are applicable in the task-free CL setting. Meanwhile, most memory based continual

Figure 2: Overview of the (a) forgetting estimation, (b) example editing, (c) model updating in the proposed GMED method. See detailed formulations for forgetting estimation and example editing at Section 4.3.

learning algorithms are applicable to the task-free setting (Aljundi et al., 2019a;b). Memory-based CL algorithms such as Experience Replay (ER) (Robins, 1995) store a subset of examples in a fix-sized replay memory and utilize them later at training to alleviate forgetting. Recent research has studied online strategies to improve the performance gain when examples get replayed from two dimensions: in terms of *which examples to store*, and *which examples to replay*. For example, in terms of deciding *which examples to store*, Gradient based Sample Selection (GSS) (Aljundi et al., 2019b) proposes to store most diverse examples. In terms of deciding *which examples to replay*, maximally Interfering Retrieval (MIR) (Aljundi et al., 2019a) select examples with the largest estimated forgetting. In particular, a task-aware approach, Hindsight Anchor Learning (HAL) (Chaudhry et al., 2020), shares the same assumption that forgettable examples should be prioritized more. However, HAL only applies to task-aware settings and requires extra memory storage to keep track of the learned anchors. Figure 1 shows a categorization of memory-based task-free continual learning.

## 3    PRELIMINARIES

In this section we first present the problem formulation of task-free continual learning and then introduce preliminaries on memory-based continual learning methods.

### 3.1    PROBLEM FORMULATION

In task-free continual learning, we consider a (potentially infinite) stream of data examples $D$, which have a non-stationary data distribution, *i.e.*, the data distribution $P(x, y)$ over time. At each time step $t$, the model receives a single or a mini batch of labeled examples $(x_t, y_t)$ from the data stream $D$. For simplicity, here we assume that example $(x_t, y_t)$ from $D$ is generated by: first sampling a *latent* "task" $z \sim P(z; t)$, followed by sampling a data example from a joint data distribution $P(x, y|z; t)$ that is conditioned on task $z$, *i.e.*, $(x_t, y_t) \sim P(x, y|z; t)$. Here $P(z; t)$ is non-i.i.d and time-dependent. Similarly, $P(x, y|z; t)$ also changes over time.

The goal of task-free online continual learning is to seek a classification model $f(x; \theta)$, parameterized by $\theta$, over new example(s) $(x, y)$ from the data stream $D$ that minimizes a predefined loss $\ell(x, y; \theta)$ while not increasing the loss on previously seen examples. This capability is evaluated by testing the model over a test set of all visited tasks.

### 3.2    MEMORY-BASED CL METHODS

Briefly, memory-based CL algorithms maintain a fix-sized replay memory $M$ which is used to store (subset of) previously seen examples $(x_t, y_t)$ from the stream $D$. When the memory is full, the algorithm needs to either identify a memory example $(x, y)$ to be replaced by new example, or to discard the new example it just received. Following the same setup in previous memory-based CL methods, our experiments use reservoir sampling (Vitter, 1985) to determine how the memory will be updated with new examples received from stream $D$. Every time the model receives a new example, it draws an integer $j$ between $0$ and $N$ randomly, where $N$ is the number of examples visited so far. If $j < |M|$ (*i.e.*, the memory size or budget), it replace the example at the $j$-th position in the memory with the new example; otherwise, this newly received example will be discarded. Reservoir sampling ensures at each time step each visited example is kept with an equal probability $|M|/N$.

At each time step $t$, the algorithm also needs to determine the memory examples to be used for replay. Similar to previous methods, we randomly sample one or a mini-batch of examples $(x, y)$ from the memory $M$. As an alternative replay strategy, MIR (Aljundi et al., 2019a) identifies a subset of memory examples based on a predefined optimization objective (*i.e,* perform one step of training on $(x, y)$), and then replays the selected examples.

## 4 GRADIENT BASED MEMORY EDITING

We propose Gradient based Memory Editing (GMED), a novel algorithm for updating stored memory examples in an online fashion. We state our hypothesis about which examples should be stored in Sec. 4.1. We then formulate an online optimization objective for example editing in Sec. 4.2. In Sec. 4.3, we introduce algorithmic details of GMED and its integration with MIR.

### 4.1 HYPOTHESIS FOR MEMORY EDITING

As there is no prior knowledge about the forthcoming examples in a data stream $D$, previous task-free CL methods usually impose (implicit) assumptions regarding what kinds of examples may improve model's test performance after replaying these examples. For example, GSS (Aljundi et al., 2019b) assumes that the diversity of memory examples contributes to the model performance; MIR (Aljundi et al., 2019a) and HAL (Chaudhry et al., 2020) assume that replaying examples that are likely to be "forgotten" can benefit the performance. Empirical study by Toneva et al. (2019) also shows that there are constantly forgotten examples, which benefit overall performance when they get replayed compared to other examples.

Our work is based on a similar hypothesis: *replaying examples that are likely to be forgotten by the current model helps retain its test performance*. Specifically, suppose we train the model on $D$ until a time step $T$, the "*forgetting*" measurement of an example $(x, y)$ for the model at $t$, denoted by $d_{t:T}(x, y)$, is defined as the "loss increase" at time $T$ compared to that at time $t$, shown as follows.

$$d_{t:T}(x, y) = \ell(x, y; \theta_T) - \ell(x, y; \theta_t), \tag{1}$$

where $\theta_t$ is the model parameters at the current time step $t$. A larger $d_{t:T}(x, y)$ indicates that the example suffers more forgetting at the end of training. The hypothesis is formally stated as follows.

**Hypothesis 1.** *Given a budget of $C$ examples to replay at time $t$, in order to minimize the loss over new examples from a (latent) task $k$ (e.g., in test set), an ideal strategy is to replay the most forgettable examples, denoted as $S_k$, selected from the training examples of the (latent) task $k$, denoted as $D_k$.*

Following Hypothesis 1, the set of examples $S_k$ can be obtained by solving the following optimization problem.

$$S_k = \underset{S_k \subseteq D_k, |S_k| = C}{\arg\max} \sum_{(x, y) \in S_k} d_{t:T}(x, y), \tag{2}$$

where $D_k$ is the training examples of the latent task $k$. Unfortunately, the sample selection problem in Eq. 2 cannot be solved in an online setting, even when we know task identities, because the objective function in Eq. 2 can only be evaluated at a distant future time step $T$. Therefore, approximations are necessary to enable online optimization.

### 4.2 ONLINE OPTIMIZATION FOR MEMORY EDITING

We propose an optimization problem that is tractable in an online fashion, which shares the same goal of making memory examples used for replay more likely to be forgotten. We modify Eq. 2 where we (1) relax the constraint $S_k \subseteq D_k$ and edit individual examples stored in memory instead of selecting the most forgettable examples from $D_k$ to store, and (2) estimate the forgetting measure in an online fashion.

Formally, suppose that the model is at the $t$-th time step and that $(x, y)$ is an example from a certain task $k$. In order to retain the performance on test examples from the same task $k$, we propose the following objective:

$$x^* = \underset{x}{\arg\max} \quad d_{t:t+1}(x, y) - \beta \ell(x, y; \theta_t), \tag{3}$$

where $d(\cdot)$ is the forgetting defined in Eq. 1 and $\ell(x, y; \theta_t)$ is the loss of the example $(x, y)$ at the current time step $t$. $\beta$ is a trade-off hyper-parameter deciding the regularization strength. The optimal $(x^*, y)$ has the same label $y$ as the original example $(x, y)$.

Specifically, we discuss two main differences in Eq. 3 compared to Eq. 2 in the Hypothesis 1.

**Estimating Forgetting Online**. We maximize the *forgetting* of the example $(x, y)$ in the upcoming update (time step $t + 1$, noted as $d_{t:t+1}(x, y)$), instead of the forgetting when the model get evaluated, *i.e.,* $d_{t:T}(x, y)$. The former can be evaluated online efficiently without any overhead on the replay memory.

**Relaxed Constraints**. Eq. 3 do not constrain $(x, y) \in D_k$. It allows $(x^*, y)$ to be an arbitrary example in the input space without being a real example from $D_k$; the sample selection problem is posed as an optimization problem in the continuous space. Nevertheless, the editing is made conservatively, so that the edited example is still likely to be an example from the original latent task. In practice, $(x, y)$ is initialized as different input examples, and we perform only one or a few gradient updates each time it is drawn from memory for replay. We also add a regularization term $\beta\ell(x, y; \theta_t)$ to discourage the loss increase on the example.

The objective in Eq. 3 is differientiable with respect to $x$, allowing us to update $x$ with gradient ascent. In the rest of this section, we introduce algorithmic details of GMED.

### 4.3 THE GMED ALGORITHM

We start by introducing the algorithm of building GMED upon the ER (Experience Replay) idea. It introduces an additional "editing" step before replaying examples drawn from the memory.

We assume at time step $t$ the model receives a stream example $(x^{(D)}, y^{(D)})$ from the training stream $D$, and randomly draws a memory example $(x, y)$ from the memory $M$. We first compute the forgetting (*i.e.,* loss increase) on the memory example $(x, y)$ when the model performs one gradient update on parameters with the stream example $(x^{(D)}, y^{(D)})$.

$$\theta'_t = \theta_t - \nabla_\theta \ell(x^{(D)}, y^{(D)}; \theta_t); \quad (4)$$

$$d_{t:t+1}(x, y) = \ell(x, y; \theta'_t) - \ell(x, y; \theta_t), \quad (5)$$

where $\theta_t$ and $\theta'_t$ are model parameters before and after the gradient update respectively. Figure 2(a) visualize the steps to compute forgetting.

Following the optimization objective proposed in Eq. 2, we perform a gradient update on $x$ to increase its forgetting, while using a regularization term to discourage the loss increase on the example at the current time step.

$$x' = x + \alpha\nabla_x[d_{t:t+1}(x, y) - \beta\ell(x, y; \theta_t)], \quad (6)$$

where $\alpha$ is a hyperparameter for the stride of the update. Figure 2(b) visualize the editing step.

**Algorithm 1:** ER with Memory Editing

**Input:** learning rate $\tau$, edit stride $\alpha$, regularization strength $\beta$, model parameters $\theta$;

**Receives**: stream example $(x^{(D)}, y^{(D)})$;

**Initialize**: replay memory $M$;

**for** $t = 1$ *to* $T$ **do**

    $(x, y) \sim M$;
    $\ell_{\text{before}} \leftarrow \text{loss}(x, y, \theta_t)$;
    $\ell_{\text{stream}} \leftarrow \text{loss}(x^{(D)}, y^{(D)}, \theta_t)$;
    //update model parameters with stream examples, discarded later;
    $\theta'_t \leftarrow \text{SGD}(\ell_{\text{stream}}, \theta_t, \tau)$;
    //evaluate forgetting of memory examples;
    $\ell_{\text{after}} \leftarrow \text{loss}(x, y, \theta'_t)$;
    $d \leftarrow \ell_{\text{after}} - \ell_{\text{before}}$;
    //edit memory examples;
    $x' \leftarrow x + \alpha\nabla_x(d - \beta\ell_{\text{before}})$;
    $\ell = \text{loss}((x', y) \cup (x^{(D)}, y^{(D)}), \theta_t)$;
    $\theta_{t+1} \leftarrow \text{SGD}(\ell, \theta_t, \tau)$;
    replace $(x, y)$ with $(x', y)$ in $M$;
    reservoir_update$(x^{(D)}, y^{(D)}, M)$;

**end**

The algorithm then discards the updated parameter $\theta'_t$, and updates model parameters $\theta_t$ with the updated memory example $(x', y)$ and the stream example $(x^{(D)}, y^{(D)})$, in a similar way to ER.

$$\theta_{t+1} = \theta_t - \nabla_\theta \ell(\{(x', y), (x^{(D)}, y^{(D)})\}; \theta_t). \quad (7)$$

We replace the original examples in the memory with the edited example. In this way, we continuously edit examples stored in the memory alongside training. Algorithm 1 summarize the proposed ER+GMED algorithm.

GMED is studied from a complementary direction compared to most prior approaches. Therefore, we can combine GMED with existing memory-based CL algorithms without much effort. We illustrate the point by proposing a hybrid approach of GMED and MIR. We include the details of the algorithm in the Appendix.

## 5 EXPERIMENTS

We compare the performance of GMED against state-of-the-art CL algorithms on five benchmark datasets. We introduce our experimental setup and discuss our results on comparisons with baselines and performance analysis.

### 5.1 DATASETS

We consider six public CL datasets in our experiments.

**Split / Permuted / Rotated MNIST** are constructed from the MNIST (LeCun et al., 1998) dataset of handwritten digit classification. Split MNIST (Goodfellow et al., 2013) partitions the dataset into 5 disjoint subsets by their labels as different tasks. The goal is to classify over all 10 digits when the training ends. Permuted MNIST (Goodfellow et al., 2013) applies a fixed random pixel permutation to the MNIST dataset as different tasks. The dataset consists of 10 tasks. The models classify over 10 digits without knowing the permutation applied. Rotated MNIST (Lopez-Paz & Ranzato, 2017) applies a fixed image rotation between 0 to 180 degree to the MNIST dataset. Similarly, the goal is to classify over 10 digits without knowing the rotation applied. Following Aljundi et al. (2019a), for MNIST experiments, each task consists of 1,000 training examples.

**Split CIFAR-10 and Split CIFAR- 100** (Zenke et al., 2017) are constructed by splitting the CIFAR-10 or CIFAR-100 (Krizhevsky, 2009) of image classification into 5 or 20 disjoint subsets by their labels. The model classifies over all 10 or 100 classes when the training ends.

**Split mini-ImageNet** (Aljundi et al., 2019a) splits the mini-ImageNet (Deng et al., 2009; Vinyals et al., 2016) image classification dataset into 20 disjoint subsets by their labels. Similarly, the models classify over all 100 classes.

We do not provide information about task identities or boundaries to the model at both training and test time. Under the taxonomy of the prior literature (van de Ven & Tolias, 2019), our Split MNIST, Split CIFAR-10, and Split mini-ImageNet experiments are under the class-incremental setup, while Permuted MNIST and Rotated MNIST experiments are under the domain-incremental setup.

### 5.2 COMPARED METHODS

For our methods, we report the performance of ER + GMED and MIR + GMED, where we build GMED upon ER or MIR as introduced in section 4.3. We compare with several task-free memory based continual learning methods: Experience Replay (ER) (Robins, 1995; Rolnick et al., 2019), Averaged Gradient Episodic Memory (AGEM) (Chaudhry et al., 2019a), Gradient based Sample Selection (GSS) (Aljundi et al., 2019b), Maximally Interfering Retrieval (MIR) (Aljundi et al., 2019a). We also compare with Bayesian Graident Descent (BGD) (Zeno et al., 2018) and Neural Dirichlet Process Mixture Model (CN-DPM) (Lee et al., 2020), which are regularization and model-expansion based approaches respectively. We also include Graident Episodic Memory (GEM) (Lopez-Paz & Ranzato, 2017) and Hindsight Anchor Learning (HAL) (Chaudhry et al., 2020), which are task-aware methods. We also reports the results of simple fine tuning, which performs online updates on model parameters without applying continual learning algorithms; and iid Online, where we randomly shuffle the data stream, so that the model visits an i.i.d. stream of examples; and also iid Offline, where we further allow multiple pass over the dataset. See appendix for detailed descriptions of compared methods and their implementation details.

By default, we set the size of replay memory as 10,000 for split CIFAR-100 and split mini-ImageNet, and 500 for all other datasets. We also report performance under various memory sizes. Following Chaudhry et al. (2019a), we tune the hyperparameters with only the training and validation set of first three tasks. We mostly follow the training setup of Aljundi et al. (2019a). For three MNIST datasets, we use a MLP classifier with 2 hidden layers with 400 hidden units each. For Split CIFAR-10, Split CIFAR-100 and Split mini-ImageNet datasets, we use a ResNet-18 classifier. See Appendix for the details of the hyperparameter tuning and model achitectures.

| Methods / Datasets | Split MNIST | Permuted MNIST | Rotated MNIST | Split CIFAR-10 | Split CIFAR-100 | Split mini-ImageNet |
|---|---|---|---|---|---|---|
| Fine tuning | $18.80 \pm 0.6$ | $66.34 \pm 2.6$ | $41.24 \pm 1.5$ | $18.49 \pm 0.2$ | $3.06 \pm 0.2$ | $2.84 \pm 0.4$ |
| iid online | $85.99 \pm 0.3$ | $73.58 \pm 1.5$ | $81.30 \pm 1.3$ | $62.23 \pm 1.5$ | $18.13 \pm 0.8$ | $17.53 \pm 1.6$ |
| AGEM (Chaudhry et al., 2019a) | $29.02 \pm 5.3$ | $72.17 \pm 1.5$ | $50.77 \pm 1.9$ | $18.49 \pm 0.6$ | $2.40 \pm 0.2$ | $2.92 \pm 0.3$ |
| GEM (Lopez-Paz & Ranzato, 2017) | $87.18 \pm 1.3$ | $78.23 \pm 1.2$ | $76.49 \pm 0.8$ | $20.05 \pm 1.4$ | $8.75 \pm 0.4$ | $11.27 \pm 3.4$ |
| GSS-Greedy (Aljundi et al., 2019b) | $84.16 \pm 2.6$ | $77.43 \pm 1.4$ | $73.66 \pm 1.1$ | $28.02 \pm 1.3$ | $19.53 \pm 1.3$ | $16.19 \pm 0.7$ |
| BGD (Zeno et al., 2018) | $13.54 \pm 5.1$ | $19.38 \pm 3.0$ | $77.94 \pm 0.9$ | $18.23 \pm 0.5$ | $3.11 \pm 0.2$ | $24.71 \pm 0.8$ |
| HAL (Chaudhry et al., 2020) | $77.92 \pm 4.2$ | $77.55 \pm 4.2$ | $78.48 \pm 1.5$ | $32.06 \pm 1.5$ | $21.11 \pm 1.4$ | $21.18 \pm 2.1$ |
| ER (Robins, 1995) | $80.96 \pm 2.3$ | $79.69 \pm 1.0$ | $76.95 \pm 1.7$ | $33.34 \pm 1.5$ | $20.65 \pm 1.3$ | $26.00 \pm 1.0$ |
| MIR (Aljundi et al., 2019a) | $84.88 \pm 1.7$ | $79.96 \pm 1.3$ | $78.30 \pm 1.0$ | $34.47 \pm 2.0$ | $20.18 \pm 1.7$ | $25.01 \pm 1.3$ |
| ER + GMED | $82.68^{**} \pm 2.1$ | $79.70 \pm 1.1$ | $77.89^{*} \pm 0.9$ | $35.01^{*} \pm 1.5$ | $20.87 \pm 1.4$ | $\mathbf{27.79^{**} \pm 0.7}$ |
| MIR + GMED | $\mathbf{87.86^{**} \pm 1.1}$ | $\mathbf{80.11^{*} \pm 1.2}$ | $\mathbf{79.16^{**} \pm 0.9}$ | $\mathbf{35.54 \pm 1.9}$ | $\mathbf{21.49^{*} \pm 0.6}$ | $26.29^{*} \pm 1.2$ |
| iid offline (upper bound) | $93.87 \pm 0.5$ | $87.40 \pm 1.1$ | $91.38 \pm 0.7$ | $75.17 \pm 0.7$ | $41.45 \pm 0.9$ | $36.54 \pm 1.4$ |

Table 1: Mean and standard deviation of final accuracy(%) in 10 runs. For Split mini-ImageNet and Split CIFAR-100 datasets, we set the memory size to 10,000 examples; we use 500 for other datasets. * and ** indicate significant improvement over the counterparts without GMED with p-values less than 0.05 and $10^{-3}$ respectively in single-tailed paired t-tests.

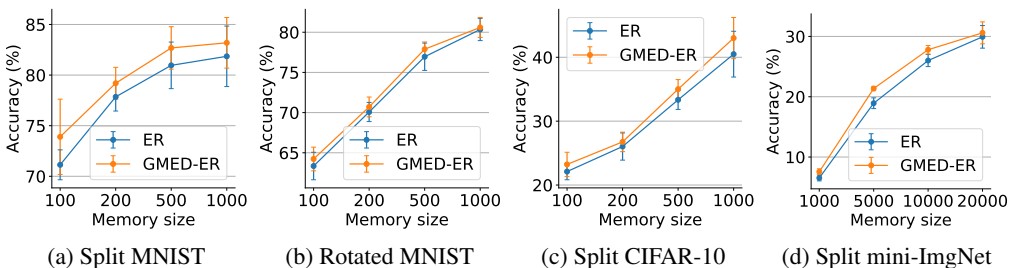

| (a) Split MNIST | (b) Rotated MNIST | (c) Split CIFAR-10 | (d) Split mini-ImgNet |
|---|---|---|---|

Figure 3: Performance of ER and GMED-ER in different memory sizes. For mini-ImageNet dataset, we use memory sizes of 1,000, 5,000, 10,000, and 20,000 examples; for other datasets, we use 100, 200, 500, and 1,000.

## 5.3 RESULTS AND PERFORMANCE ANALYSIS

We report the final accuracy achieved by different methods in Table 1. We summarize the following findings.

**Overall Performance.** From the results, we see MIR + GMED achieves best performance among memory based continual learning algorithms. The improvement of GMED differs by datasets. ER+GMED clearly improves over ER by an absolute margin of 1.72%, 1.67%, and 1.79% accuracy respectively in Split MNIST, Split CIFAR-10, and Split mini-ImageNet datasets. On Rotated MNIST and Split CIFAR-100 the improvement is less significant, with an absolute accuracy improvement of 0.94% and 0.22%, while we do not see a meaningful improvement on Permuted MNIST dataset. MIR+GMED improves performance on all the datasets. The improvement is clear on Split MNIST, Split CIFAR-10, Split CIFAR-100, and Split mini-ImageNet with an absolute accuracy improvement of 2.98%, 1.07%, 1.31%.

**Performance Under Various Memory Sizes.** Figure 3 shows the performance under various memory sizes. We see in Split MNIST, Rotated MNIST, Split CIFAR-10 and Split mini-ImageNet, the improvement of ER+GMED over ER is consistent under various memory sizes.

| Methods / Datasets | Split MNIST | Permuted MNIST | Rotated MNIST | Split CIFAR-10 | Split CIFAR-100 | Split mini-ImageNet |
|---|---|---|---|---|---|---|
| **ER + GMED** | $82.68 \pm 2.1$ | $79.70 \pm 1.1$ | $77.89 \pm 0.9$ | $35.01 \pm 1.5$ | $20.87 \pm 1.4$ | $27.79 \pm 0.7$ |
| **ER + Random Edit** | $81.04 \pm 2.1$ | $78.99 \pm 0.5$ | $77.59 \pm 1.0$ | $32.26 \pm 1.9$ | $20.73 \pm 1.1$ | $26.23 \pm 2.1$ |
| **MIR + GMED** | $87.86 \pm 1.1$ | $80.11 \pm 1.2$ | $79.16 \pm 0.9$ | $35.54 \pm 1.9$ | $21.49 \pm 0.6$ | $26.29 \pm 1.2$ |
| **MIR + Random Edit** | $84.76 \pm 1.4$ | $79.76 \pm 0.6$ | $78.19 \pm 1.0$ | $35.39 \pm 3.0$ | $19.77 \pm 1.1$ | $24.86 \pm 0.7$ |

Table 2: Performance when we updates memory examples to a random direction (Random Edit), compared to ER+GMED and MIR+GMED.

| Method | Split MNIST | | Split CIFAR-10 | | Split CIFAR-100 | |
|---|---|---|---|---|---|---|
| | Acc. | #. Mem | Acc. | #. Mem | Acc. | #. Mem |
| **ER** | 92.67 | 2,581 | 62.96 | 6,024 | 21.79 | 21,295 |
| **ER+GMED** | 94.16 | 2,581 | 63.28 | 6,024 | 22.12 | 21,295 |
| **CN-DPM** | 93.23 | 500 + Gen. | 45.21 | 1,000 + Gen. | 20.10 | 1,000 + Gen. |

Table 3: Comparison with CN-DPM under the same memory overhead. The overhead is the size of the replay memory plus the extra model components (e.g. a generator (Gen.)), shown in the equivalent number of memory examples (#. Mem).

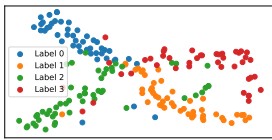

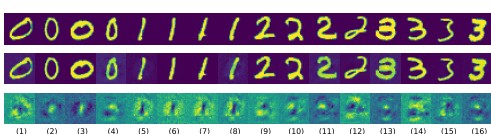

Figure 4: A t-SNE visualization of the editing performed on data examples. We use labels from the first two tasks in Split MNIST.

Figure 5: Visualization of the editing on examples in Split MNIST. The first two rows show examples before and after editing, and the third row shows the differences.

**Comparison with Random Editing**. As an ablation study, we show the result of a random editing baseline in Table 2. The baseline edits memory examples to a random direction with a fix stride, tuned in a similar way as GMED. We see GMED outperforms random editing in all cases. We further notice ER+Random Edit outperforms ER on split CIFAR-10 and CIFAR-100. We conjecture the reason is that the random editing alleviates the overfitting to memory examples.

**Comparison with Model Expansion Approach (CN-DPM)**. Non-memory based continual learning approaches introduce extra overhead in storing model parameters — for example, CN-DPM introduces extra overhead by employing a generative model component and a short-term memory (STM) in addition to the classifier. Following Hsu et al. (2018), we set the memory size for GMED so that two methods introduces the same amount of the overhead. Table 3 show the results of ER, ER+GMED and the reported results of CN-DPM. We see ER+GMED outperforms CN-DPM.

## 5.4 CASE STUDY AND DISCUSSION

**Visualization of Edited Examples.** Figure 5 visualize the editing on memory examples. We show examples from first two task (0/1, 2/3) in the Split MNIST dataset. The first and second rows show the original and edited examples, noted as $x_{\text{before}}$ and $x_{\text{after}}$. The third row shows the difference between two $\Delta x = x_{\text{after}} - x_{\text{before}}$. We see no significant visual differences between original and edited examples. However, by looking at the difference $\Delta x$, we see there are examples whose contours get exaggerated, *e.g.*, examples 1 and 12, and some get blurred, *e.g.*, examples 2, 3, 5, and 6. Intuitively, to make an ambiguous example more forgettable, the editing should exaggerate its features; while to make a typical example more forgettable, the editing should blur its features. Our visualizations align with the intuition above: examples 1 and 12 are not typically written digits, while examples like 2, 3, 5, and 6 are typical.

**Visualization of Editing Directions**. In Figure 4, we show the t-SNE Maaten & Hinton (2008) visualization of the editing vector $\Delta x = x_{\text{after}} - x_{\text{before}}$ for examples from first 2 tasks in Split MNIST. We see the editing vectors cluster by the labels of the examples. It implies the editing performed is correlated with the labels and is clearly not random.

## 6 CONCLUSION

In this paper, we propose Gradient based Memory Editing for task-free continual learning. The approach estimates forgetting of stored examples online and edit them so that they are more likely to be forgotten in upcoming updates. Experiments on benchmark datasets show our method can be combined with existing approaches to significantly improve over baselines on several benchmark

datasets. Our analysis further show the method is robust under various memory sizes, and outperforms alternative editing methods.

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

## A    DETAILS OF COMPARED METHODS

We included detailed descriptions, and implementation details of some selected baselines in this section.

• **Experience Replay (ER)** Robins (1995); Rolnick et al. (2019) stores examples in a fix-sized memory for future replay. We use reservoir sampling to decide which examples to store and replace. Following prior works Aljundi et al. (2018; 2019a); Chaudhry et al. (2020), at each time step we draw the same number of examples as the batch size from the memory to replay, which are both set to 10. The algorithm applies to the task-free scenario.

• **Gradient Episodic Memory (GEM)** Lopez-Paz & Ranzato (2017) also stores examples in a memory. Before each model parameter update, GEM project gradients of model parameters so that the update does not incur loss increase on any previous task. The approach is not task-free.

• **Averaged Gradient Episodic Memory (AGEM)** Chaudhry et al. (2019a) prevents the average loss increase on a randomly drawn subsets of examples from the memory. We draw 256 examples to compute the regularization at each iteration. The approach is task-free.

• **Bayesian Gradient Descent (BGD)** Zeno et al. (2018) is a regularization-based continual learning algorithm. It adjust learning rate for parameters by estimating their certainty, which notes for their importance to previous data. The approach is task-free.

• **Gradient based Sample Selection (GSS)** Aljundi et al. (2019b) builds upon ER by encouraging the diversity of stored examples. We use GSS-Greedy, which is the best performing variant in the paper. The approach is task-free.

• **Hindsight Anchor Learning (HAL)** Chaudhry et al. (2020) learns an pseudo "anchor" example per task per class in addition to the replay memory by maximizing its estimated forgetting, and tries to fix model outputs on the anchors at training. However, unlike GMED, they estimate forgetting with loss increase on examples when the model train for a pass on the replay memory (and thus forgetting is estimated with "hindsight"). The approach is not task-free.

• **Maximally Interfering Retrieval (MIR)** Aljundi et al. (2019a) improves ER by selecting top forgettable examples from the memory for replay. Following the official implementation, we evaluate forgetting on a candidate set of 25 examples for mini-ImageNet dataset, and 50 examples for others. While the approach is task-free, the official implementation filter out memory examples that belong to the same task as the current data stream, which assumes knowledge about tasks boundaries. We remove this operation to adapt the method to the task-free setup. Therefore, our results are not directly comparable to the official results.

- **Neural Dirichlet Process Model for Continual Learning (CN-DPM)** Lee et al. (2020) is a task-free model-expansion based continual learning algorithm. We report the official results in the paper. In the comparison study between ER/ER+GMED with CN-DPM, for the base model in ER/ER+GMED, we use the full expanded model in CN-DPM (*i.e.*, the model architecture when the training ends in CN-DPM). We use the same optimizer and the learning rate as CN-DPM in this set of experiments.

## B  HYPERPARAMETER SETUP

We use a batch size of 10 throughout the experiment. We use SGD optimizer with a learning rate of 0.05 for MNIST datasets, 0.1 for Split CIFAR-10 and Split mini-ImageNet datasets, and 0.03 for the Split CIFAR-100 dataset. We perform three steps of model parameter updates for each example we visit in the Split mini-ImageNet dataset, following Aljundi et al. (2019a), and one step for others.

| Dataset / Hyper-param | Editing stride $\alpha$ | Regularization strength $\beta$ |
|---|---|---|
| Split MNIST | 5.0 | 0.01 |
| Permuted MNIST | 0.05 | 0.001 |
| Rotated MNIST | 1.0 | 0.01 |
| Split CIFAR-10 | 0.05 | 0.001 |
| Split CIFAR-100 | 0.05 | 0.001 |
| Split mini-ImageNet | 1.0 | 0.1 |

Table 4: Hyperparamters of the editing stride and the regularization strength selected for GMED.

GMED introduces two hyperparameters: the stride of the editing $\alpha$, and the regularization strength $\beta$. As we assume no access to the full data stream in the online learning setup, we cannot select hyperparameters according to validation performance after training on the full stream. Therefore, We tune the hyperparameters with only the training and validation set of first three tasks, following Chaudhry et al. (2019a). The models are trained until convergence before they proceed to the next task. The tasks used for hyperparameter search are included for reporting final accuracy, following Ebrahimi et al. (2020). We perform a grid search over all combinations of $\alpha$ and $\beta$ and select the one with the best validation performance on the first three tasks. We select $\alpha$ from $[0.01, 0.05, 0.1, 0.5, 1.0, 5.0, 10.0]$, and select $\beta$ from $[0, 10^{-3}, 10^{-2}, 10^{-1}, 1]$. We tune hyperparameters for ER-GMED and apply the same hyperparameters on MIR+GMED. Table 4 show the optimal hyperparameters selected for each dataset.

## C  IMPLEMENTATION DETAILS OF THE HYBRID METHOD OF GMED AND MIR

As mentioned in Sec. 4.3, GMED can be built upon MIR to further improve the performance. At each time step, MIR retrieves the most forgettable examples from the memory with the forgetting defined as Eq. 1. We do not edit the selected examples directly; instead, we additionally draw another random mini-batch from the memory to apply editing. The motivation is that examples drawn by MIR are already most forgettable ones; if we directly perform editing on them, we would fall into a loop that letting forgettable examples more forgettable, which is not desired.

