# OpenReview forum: "Gradient Based Memory Editing for Task-Free Continual Learning"
_ICLR.cc/2021/Conference — Reject_

### Official Review · AnonReviewer2 · 2020-10-27
**Intriguing Empirical Results, But Lacking Theoretical Justification**

**Rating:** 6
**Confidence:** 4

**Review:**

## Summary

This paper proposes a task-free continual learning method called GMED that extends experience replay.
The key idea is to modify the individual data points in the replay memory to maximize the one-step forgetting at the current time step.

---

## Pros

- The idea of editing the data in a replay buffer with gradient descent is novel.
- The experiments show a significant performance gain compared to the baselines.

---

## Cons

### Lack of justification for editing memory

Given a somewhat arbitrary objective function, updating the data in memory by gradient descent will make the data points move away from their original distribution. Since continual learning aims to learn the data's distribution, I think modifying each data point does not align with this goal.

One possible explanation for the performance improvement is that the editing acts as a kind of data augmentation. As shown in Figure 5, the difference between the edited data and the original data is minuscule. Since the performance would drop if the editing is significant, the magnitude of updates should be small such that the data do not deviate too far from the original distribution.

From this point of view, I think it is critical to show that GMED is better than other data augmentation tricks.
The authors already performed an ablation study showing that GMED is better than adding random noise to the data.
However, there is no much detail about the experiment to verify its validity. Since it is an important experiment, I strongly recommend the authors to provide enough details for reproducing.
Also, I would like to see a comparison with other standard data augmentation methods.

### Confusing motivation and writing

The fundamental hypothesis of this paper is:
> replaying examples that are likely to be forgotten by the current model helps retain its test performance.

However, the final algorithm is far from this hypothesis. As I mentioned previously, I think it is closer to a data augmentation algorithm.

### Unreasonable experimental setting

For the Split CIFAR-100 and Split mini-ImageNet experiments, the authors use a memory size of 10,000. However, the size of the whole training set is 50,000, which means the memory can store up to 20% of the whole data. In my opinion, the current trend is to keep the memory size under 1-2% of the whole dataset, which corresponds to 500-1000 in the case of CIFAR100. Note that the authors also follow this trend for MNIST and CIFAR-10. With 20% of the dataset, I suspect that even i.i.d. training on the memory would not differ much from i.i.d. training on the whole dataset.

Also, I think it is more constructive to compare models under a fixed number of examples in memory, not the number of bytes as in Table 3. While trying to equalize the memory usage of ER(+GMED) and CN-DPM, the number of examples in memory became unreasonably large for ER(+GMED) (over 40% of the whole data).

The brutal fact is that the current level of continual learning is far from practical. At present, most CL papers work on toy problems like CIFAR-100. At this level, I think comparing the actual memory usage is not meaningful. In my opinion, what the community should look for is an efficient algorithm that can scale to real-world problems. For this reason, I think it is more meaningful to constrain the number of examples in memory. Also, note that if we scale up the image size, the size of image data can be more dominant than CNN's parameter size.

---

## Overall evaluation

The proposed method does not have any theoretical justification, and several settings in the experiments are unrealistic. Nonetheless, since the experiments show significant improvements over baselines, I think it can be an effective technique. Unfortunately, at the current state of the paper, I cannot exclude the possibility that simpler data augmentation tricks on the replay memory may outperform the proposed method. Although I currently vote for rejection, I am willing to raise the score if the authors can offer enough evidence that GMED is better than trivial data augmentation tricks.

---

## Post-rebuttal

During the rebuttal period, the experiments largely improved, resolving the majority of my concerns. I raise my rating accordingly. I expect the authors to reflect other suggestions as well in the final version. Especially, R1 raised a serious concern about the similarity between GMED and GEN/AE-MIR. A proper comparison should be included in the final version.

---

> ### Author Response · Authors · 2020-11-20
> **Response to Reviewer #2**
>
> Thanks for your insightful comments!
>
> **Q1 Lack of justification for editing memory. Given a somewhat arbitrary objective function, updating the data in memory by gradient descent will make the data points move away from their original distribution. Since continual learning aims to learn the data's distribution, I think modifying each data point does not align with this goal.**
>
> According to Hypothesis 1, it is true that we expect data points to stay within their original distribution while maximizing their estimated forgetting. To achieve this, we have introduced regularization terms, and performed very conservative updates on examples (e.g., only one example at each time step). Our approach is reasonably approximating our hypothesis; even though it may not be perfect but it is one good solution that could improve performance.
>
> **Q2. Comparison between GMED and other data augmentation tricks & The fundamental hypothesis of this paper is: replaying examples that are likely to be forgotten by the current model helps retain its test performance. However, the final algorithm is far from this hypothesis. I think it is closer to a data augmentation algorithm.**
>
> Thanks for your thoughtful comments. We agree that data augmentation could be a reason behind GMED’s improvement - the random editing baseline reported in our paper improves over counterparts without editing on split CIFAR-10. In our follow-up experiments, we verified that GMED outperforms other variants of data-augmentation approaches and the following alternative editing objectives.
>
> - Editing examples to the opposite direction as GMED (Flipped GMED)
> - Creating adversarial examples during training with PGD-like editing objective [2]
>
> The results are shown in the table below (we select four datasets where there are clear differences in results between different memory-based algorithms). We re-run ER, ER-GMED methods with a new and the same set of 5 random seeds. We see GMED outperforms alternative editing methods.
>
> Method/Dataset | Split MNIST | Rotated MNIST | Split CIFAR-10 | Split mini-ImageNet
> -------- | ------- | ------- | ------- | ------
> ER+GMED | 82.82 | 77.76 | 36.21 | 28.19
> ER+Flipped GMED | 74.82 | 77.41 | 32.67 | 27.96
> ER+Adversarial examples | 75.02 | 72.75 | 33.64 | 25.07
>
>  **Q3. One possible explanation for the performance improvement is that the editing acts as a kind of data augmentation. As shown in Figure 5, the difference between the edited data and the original data is minuscule. Since the performance would drop if the editing is significant, the magnitude of updates should be small such that the data do not deviate too far from the original distribution.**
>
> We further provide clarification regarding the reasoning of the reviewer’s reasoning above. We agree we should not make significant changes to memory examples. GMED takes measures to prevent edited examples from deviating too much from original examples (e.g. regularization term in Eq. 3). However, The small editing performed also does not imply the algorithm is similar to data augmentation. We argue that a small but principled change in input examples may affect model behaviors in a significant way. For example, in the literature that studies adversarial attacks, adversarial examples may effectively flip the model's predictions, even if they look visually the same as original examples.
>
> **Q4. Questions about memory sizes**
>
> For split mini-Imagenet, we borrowed the experimental setups from [1]. We applied the same memory size on split CIFAR-100 given the similarity of the stream length, the total number of tasks, and the total number of classes. We agree that memory-based continual learning algorithms should try to reduce the size of the memory. However, we find that under 500~1000 memory examples, the performance on these two datasets is too low (below 10% accuracy) to draw meaningful conclusions.
>
> [1] Aljundi, Rahaf, et al. "Online continual learning with maximal interfered retrieval." Advances in Neural Information Processing Systems. 2019.
> [2] Madry, Aleksander, et al. "Towards deep learning models resistant to adversarial attacks." ICLR 2018

---

> > ### Comment · AnonReviewer2 · 2020-11-23
> > **Re: Response to Reviewer #2**
> >
> > **Nice explanation for the small amount of memory update**
> >
> > I thank the authors for addressing my concerns. It is convincing that even a small update on the memory may have positive effects as an adversarial attack does in the opposite way. Although there is still not much theoretical justification, I think this argument is sufficient to support that the proposed method can be an effective heuristic. Additionally, I recommend the authors to emphasize that the update is quite small. Until I checked the experiments, I was confused about how editing did not harm the performance since the word "editing" sounds like a significant modification.
> >
> > **Insufficient experimental evidence**
> >
> > However, I am still not convinced that the experimental results are enough. Since I think of GMED as a kind of data augmentation trick, one of my main concerns is that the proposed method may perform worse than standard data augmentation techniques. Here, the *standard* includes random crop, horizontal flipping, random rotation, adding noise, etc. Since the authors did not provide experiments with such techniques, and I was also concerned about the validity of the memory size, I ran some quick tests myself.
> >
> > I took a plain, iid ResNet-18 classifier code and modified the data pipeline to use only the specified number of data points. Following the memory size of the paper, I tested 500 and 10k for the number of data points. I call these experiments *mem-iid*.  I think it is reasonable to expect a replay-based CL algorithm to perform better than mem-iid. I tested mem-iid with/without some of the standard data augmentation tricks: random crop/rotation and horizontal flip. The results were quite surprising:
> >
> > | Method | CIFAR-10 | CIFAR-100 |
> > |---|---|---|
> > | ER | 33.34 | 20.65 |
> > | MIR | 34.47 | 20.18 |
> > | ER + GMED | 35.01 | 20.87 |
> > | MIR + GMED | 35.54 | 21.49 |
> > | iid offline (from paper) | 75.17 | 41.45 |
> > | mem-iid (500) | 29.89 | 6.10 |
> > | mem-iid (500) + std. aug. | 46.79 | 10.90 |
> > | mem-iid (10k) | - | 31.72 |
> > | mem-iid (10k) + std. aug.| - | 54.71 |
> > | mem-iid (50k) | 87.60 | 60.19 |
> > | mem-iid (50k) + std. aug. | 92.87 | 73.51 |
> >
> > The first five rows are excerpted from table 1 of the paper, and the numbers next to "mem-iid" indicate the number of training data. Since the size of the CIFAR10/100 dataset is 50k, `mem-iid (50k)` is equivalent to iid offline. Please note that I did not tune any hyperparameters from the original code and ran each test only once. There are several points to note:
> > - The memory size of 10k is unreasonably large. Even without any data augmentation, mem-iid easily surpasses the reported scores in CIFAR100.
> > - Although it is not a strictly fair comparison, the standard data augmentation seems to be far more effective than GMED. The standard augmentation boosts the performance significantly, while GMED improves the accuracy by only about 1 percentage point.
> > - `iid offline`'s score is too low compared to my implementation (`mem-iid (50k)`). For example, I observed that the accuracy of CIFAR10 easily goes over 80% only after a few epochs even without data augmentation. This suggests that the baselines might be poorly tuned.
> >
> > Considering the experimental results described above, I think the proposed method would not be useful.

---

> > > ### Author Response · Authors · 2020-11-25
> > > **Response to Reviewer #2**
> > >
> > > We thank the reviewer for the careful check into the experiments! Here are our responses to new questions, with some new experimental results.
> > >
> > > **The gap between ER and iid-training on the memory is close**
> > >
> > > We believe it is a question towards the general practice of memory-based continual learning . However, memory-based continual learning approaches are still useful in the online continual learning (OCL) setup. In OCL, a model is trained over an online stream of data and may be queried *at any time step*; the practice above, however, requires a new model to be trained each time the model is queried for testing. Therefore, we argue memory based continual learning algorithms should not be compared against the approach above.
> > >
> > > Besides, the good performance of iid-training on memory could be a gain from longer epochs of training until convergence. However, in OCL, the model visits a single pass of the online stream, and there is no "validation set" to determine convergence. Retraining over the whole task also does not adhere to the OCL setup.
> > >
> > > **Although it is not a strictly fair comparison, the standard data augmentation seems to be far more effective than GMED. The standard augmentation boosts the performance significantly, while GMED improves the accuracy by only about 1 percentage point.**
> > >
> > > We very much thank the reviewer for verifying the benefit of data augmentation with experiments in an iid-offline setting. We also performed our own experiments in the OCL setup. Given that the main concern is "whether editing on memory acts worse than standard data augmentation", we design the following experiment: each time we draw a mini-batch of examples from the memory for replay, we transform the examples with standard data augmentation techniques (random crop, random horizontal flip, random rotation). We feed both the original mini-batch and the transformed mini-batch to the model for replay.
> > >
> > > We see data augmentation has significantly improved performance over the reported results for ER; however, GMED can be built upon data augmentation to further improve the performance; also, the improvement is not diminished because of the data augmentation applied. We run new experiments for 5 runs with the same set of random seeds. We used the same set of hyperparameters as before.
> > >
> > > Method | Acc.
> > > ---- | ----
> > > ER (reported) | 33.34
> > > ER + GMED (reported) | 35.01
> > > ER + data aug. | 46.13
> > > ER + GMED + data aug. | 49.38*
> > >
> > > *: Better than ER + data aug. with a p-value < 0.05 in a single-tailed paired t-test
> > >
> > > Previously, we used a memory size of 10,000 for split CIFAR because the memory below this number yields very weak classifiers (e.g. with a memory size of 1000, the accuracy is around 8%). After applying data augmentation, we find the performance has improved, and thus we also report the results on CIFAR-100 with a memory size of 1,000.
> > >
> > > Method | Acc.
> > > ---- | ----
> > > ER + data aug. | 14.16
> > > ER + GMED + data aug. | 14.87
> > >
> > > We thank the reviewer for pointing out the huge benefit of data augmentation in OCL; previously, given that data augmentation is a generally applicable approach for memory-based approaches,  for consistence with previous works, we did not apply data augmentation for any approaches. However, given such a significant gain of data augmentation, it may be preferable to evaluate the effectiveness of algorithms on the basis where data augmentation is applied. We will report results with standard data augmentation for other methods when they are ready.
> > >
> > > **iid-offline score is too low**
> > >
> > > Thanks for pointing out the issue. We used the settings from [1] (e.g. family of optimizers) to run iid-offline experiments (for Split MNIST, Split-CIFAR-10, and Split mini-Imagenet), and our reported number of baselines are mostly consistent with [1]. We will re-run iid-offline experiments on Split-CIFAR 100.
> > >
> > >
> > > [1] Aljundi, Rahaf, et al. "Online continual learning with maximal interfered retrieval." Advances in Neural Information Processing Systems. 2019

---

### Official Review · AnonReviewer1 · 2020-10-28
**ok paper, but gradient-based memory editing for CL was already proposed**

**Rating:** 3
**Confidence:** 5

**Review:**

**Summary**

The authors propose a gradient-based memory editing scheme for replaying samples that are undergoing the most forgetting, coined GMED. They also propose a hybrid method with MIR [1]. An extensive evaluation on standard benchmarks shows some improvements throughout.

**Concerns**

My main concern is that the authors seem to have completely missed that gradient-based memory editing has been proposed in [1]. Specifically, in their GEN-MIR and AE-ER methods, gradient-based optimization is performed on some latent codes of the data to maximize the "forgetting" measurement.

GEN-MIR seems superior to GMED as it can perform multiple gradient-based edits on the data to achieve a significantly different replay sample. GMED can't because it re-uses the initial label as the label of the editable sample. More GMED edits will thus increase the probability that the initial label is wrong. I think this is why the authors only do one edit. You can see, however, that one edit doesn't seem to change the replay data, as exemplified in Figure 5. [1] also proposes AE-MIR when $p(x)$ is difficult to model online. Furthermore, I would argue that gradient-based editing is more sensible in latent space (GEN-MIR and AE-MIR) than in input space (GMED) because it will be much easier to stay on the data manifold. It is also much more computationally efficient.

All of section 4.1 and most of section 4.2, including "estimating forgetting online" can be found in [1]. Algorithm 1 in section 4.3 is also astonishingly similar to the ones in MIR.

I think this work is too incremental to merit a publication at a top conference like ICLR. Thus, I encourage the authors to either submit to a lower-tier conference or further develop the methodology.

**minor concerns**

- The related work seems to be missing a branch of task-free CL, namely Continual-Meta Learning.

- no need to re-explain Reservoir Sampling

- typos: a lot of hyphens are missing. e.g. *gradient-based* memory editing


__________

**POST-REBUTTAL**

Sadly, I'm decreasing my score a notch because the authors lack a deep understanding of MIR [1] that would make it evident that GMED is *much* closer to the three methods proposed in MIR. Specifically, the authors have responded to my concern about the lack of novelty:

    In GEN-MIR, the classifier and the generator separately retrieve most forgettable examples for themselves. The generator is indeed optimized for “maximizing the forgetting”, but the “forgetting” here is measured for the generator - i.e., the generator retrieves most forgettable examples for the generator itself. The approach does not learn a generator that can “generate examples that are more forgettable for the classifier”; instead, feeding more forgettable examples in GEN-MIR aims at reducing the forgetting of the generator.

This is simply not true. If you take a look at Equation 2 in MIR, you will see that the generator is generating forgettable examples **for the classifier**. It also uses it for itself, see Equation 3. GEN-MIR is thus a gradient-based memory editing for CL.

    AE-MIR (...) It is a hybrid approach of example compression and ER-MIR. There is no online optimization towards more forgettable examples for the classifier like GMED

Again, just like GEN-MIR, AE-MIR uses gradient-based memory editing for CL for the classifier.

In GMED lies somewhere between ER-MIR and [GEN-MIR, AE-MIR]. My guess is that the MIR's author didn't propose the GMED method because it doesn't make a lot of sense to update edit a sample **and not edit its label**.

Here is an intuition on the behavior of each method: let's say your models sequentially learning to visually classify objects. The model is now learning about zebras and it's causing some interference on the horse's representation.

- ER-MIR will search in its buffer and retrieve a horse for the classifier to do replay on.
- GEN-MIR will search inside the latent space of a generative model to find generated horses for the classifier to do replay on.
- AE-MIR will search the latent space of an autoencoder to retrieve past horses that appeared in the data stream for the classifier to do replay on.
- GMED randomly samples some data in the buffer, e.g. a car, and takes one gradient update on the car such that it resembles more a horse. Then the classifier is fed that modified image (i.e. x) as well as the unchanged horse label (i.e. y)

The empirical section shows us that one needs to add MIR to GMED to obtain the best results. This comes as no surprise. Combining methods with each other and increasing computing needs and/or replay will give you a better performance on forgetting.




_________


[1] Rahaf Aljundi, Lucas Caccia, Eugene Belilovsky, Massimo Caccia, Min Lin, Laurent Charlin, andTinne Tuytelaars.  Online continual learning with maximally interfered retrieval.  In NeurIPS 2019.

---

> ### Author Response · Authors · 2020-11-20
> **Response to Reviewer 1 (2/2)**
>
> **Q2: GMED edits will increase the probability that the initial label is wrong. I think this is why authors only do one edit**
>
> It is possible but if we edit examples too drastically. However, by making editing rather conservative, we have achieved better performance on several datasets.
>
> Our editing mechanism is motivated by a well-studied hypothesis (forgettable examples are more helpful). The approach is reasonably approximating our hypothesis; even though it may not be perfect but it is one good enough solution to improve performance on several datasets.
>
> **Q3: You can see, however, that one edit doesn't seem to change the replay data, as exemplified in Figure 5**
>
> Indeed, our editing will not bring significant changes in the input space, which is a preferable consequence of the proposed algorithm, because it confirms examples stay in original distributions and the original label is likely to be correct for the edited example. However, just like it is possible to create “adversarial examples” that could flip model’s predictions with indistinguishable differences by humans, a small change on input examples may affect model behaviors in a significant way. Our experiments confirm such small updates are useful.
>
>
> **Q4: Does GMED work when p(x) is difficult to model?**
>
> In our experiments, we show that in the most challenging dataset, mini-ImageNet, GMED brings statistically significant improvements. Besides, on Split-CIFAR 10, which is believed to be a challenging dataset for generative replay based methods [1], GMED also brings improvements. Therefore, even when p(x) is moderately difficult, GMED still works well.
>
> **Q5: All of section 4.1 and most of section 4.2, including "estimating forgetting online" can be found in [1]. Algorithm 1 in section 4.3 is also astonishingly similar to the ones in MIR.**
>
> Section 4.1 states our hypothesis of how memory editing should be performed. We cited [1], and also another methodological paper [2], and an empirical study [3] to support that “examples that are likely to be forgotten should be prioritized for replay” is a common hypothesis applied in previous methodology papers and with empirical justification.
>
> Thanks for pointing out the issue of section 4.2. It seems the current writing has caused some confusion. We will update the draft and add references right after our introduction about how to estimate forgetting online.
>
> Since MIR and GMED apply the same hypothesis (examples that are likely to be forgotten should be prioritized for replay), it is natural that the algorithms share some steps. However, we argue that the example editing in GMED is orthogonal to example retrieval in MIR, and they can be even combined to achieve better results (the reported MIR-GMED method).
>
> **Q6: Missing comparison & references of a branch of task-free CL, namely Continual Meta learning**
>
> Thanks for pointing out. The proposed GMED method is focused on improving existing memory-based methods. We will add references to these methods in our related works section.
>
> [1] Aljundi, Rahaf, et al. "Online continual learning with maximal interfered retrieval." Advances in Neural Information Processing Systems. 2019.
> [2] Chaudhry, Arslan, et al. "Using hindsight to anchor past knowledge in continual learning." arXiv preprint arXiv:2002.08165
> [3] Toneva, Mariya, et al. "An empirical study of example forgetting during deep neural network learning." ICLR 2019.

---

> ### Author Response · Authors · 2020-11-20
> **Response to Reviewer 1 (1/2)**
>
> Thanks for your thoughtful comments!
>
> **Q1: Has the gradient-based memory editing already been proposed in [1]? What are the differences between GMED and GEN-MIR / AE-MIR**
>
> We appreciate the reviewer's careful check into [1]. However, we would like to point out there are some misunderstandings of GEN-MIR and AE-MIR in the review, which may have caused the impression that GMED and GEN-MIR / AE-MIR are similar.
>
> In GEN-MIR, the classifier and the generator separately retrieve most forgettable examples for themselves. The generator is indeed optimized for “maximizing the forgetting”, but the “forgetting” here is measured for the generator -  i.e., the generator retrieves most forgettable examples for the generator itself. The approach does not learn a generator that can “generate examples that are more forgettable for the classifier”; instead, feeding more forgettable examples in GEN-MIR aims at reducing the forgetting of the generator. In contrast, GEMD aims at  “editing examples so that they are more forgettable for the classifier. We agree that the approach above (learning to generate forgettable examples for the classifier) can be potentially an extension of GMED and we are interested in further study.
>
> We noticed the GEN-MIR method in [1], but we did not discuss it in the paper because the performance is outperformed by ER-MIR by a large margin on Split MNIST and only a small performance improvement is observed on Permuted MNIST.
>
> Method/Dataset | Split MNIST | Permuted MNIST
> -------- | -------- | --------
> ER 		|    80.96	| 79.69
> ER-GMED	|	82.68	| 79.70
> ER-MIR (task-free version) |	84.88 | 79.96
> ER-MIR-GMED |	87.86	| 80.11
> GEN-MIR (reported in [1])	|	82.1	| 80.04
>
> Furthermore, we do not see reported results for Split CIFAR10 and Split mini-ImageNet for GEN-MIR in [1], while there are reported results for ER-MIR. In fact, training a generative model on more challenging datasets compared to MNIST especially in an online (single-pass), task-free setup is challenging, which is also pointed out in [1]. GMED does not require a generative model, and therefore it is extendable to more complicated datasets (e.g. split mini-imagenet) and improves performance on these datasets.
>
>
> AE-MIR relies on an *encoder-decoder pre-trained offline* to compress input data before storing them in the memory with the encoder, and perform MIR search over them and decode them for replay. It is a hybrid approach of example compression and ER-MIR. There is no online optimization towards more forgettable examples for the classifier like GMED.

---

### Official Review · AnonReviewer4 · 2020-10-29
**An interesting method to improve memory based continual learning**

**Rating:** 7
**Confidence:** 4

**Review:**

This paper deals with continual learning. Specifically, given a stream of tasks we want to maximise performance across all tasks. Typically neural networks suffer from catastrophic forgetting which results in worse performance on tasks seen earlier in training. There are many proposed solutions to this problem. One specific set of approaches are "memory based" algorithms. Here we store some training examples in memory from the tasks seen thus far. These are then mixed in with new training data so as to encourage the model to not forget past tasks.

There are two central question for approaches of these kinds: what examples to store and how to use them. This paper details a method that is complementary to most approaches. Specifically, given a system of storing examples (here reservoir sampling) and using them, this methods deals with how best to extract performance from an example. Specifically, it "edits" the example before storing and replaying it, to make it more useful to the model.

The main motivation for the editing process is as follows: past works have shown that the most useful examples to replay are those that are most likely to be forgotten. Thus, if we take our examples in memory and update them (via gradient ascent) to be "more forgettable", then when replayed these will result in the most benefit to the model.

* One comment I have here is that the way this motivation is presented in the paper can be improved; perhaps with a more clearly written paragraph in the preliminaries. On my first reading of the introduction for example, this sentence for example, "_edits examples stored in the memory with gradient-based updates so that they are more likely to be forgotten_" sounds like you're trying to get the model to forget examples more which is the opposite of what we want (as opposed to making the example inherently more forgettable so it is better when replayed).

* The reasoning here is that because naturally forgettable examples help during replay, _making_ an example "forgettable" will naturally improve its usefulness. I can't fault this reasoning based on the results but to me it is not naturally clear that this follows. Would the authors be able to provide some intuition for what that means from an optimisation perspective?

Further, the authors propose an online metric for how much the network forgets an example (by comparing the loss at consecutive timesteps). Then when taking an example from memory, we perform a gradient update on the image to optimise this metric.

Lastly, we replay the edited example to the network and perform a gradient update based on both this edited example and the current training example. Figure 2 is very useful and well done to illustrate this.

The experiments are performed across standard tasks and show modest to good improvement over the range of datasets.

 * Two baselines that would be useful to include in the table are EWC (https://arxiv.org/pdf/1612.00796.pdf) and MbPA ((https://arxiv.org/pdf/1802.10542.pdf), the latter of which is a memory based method that also does not require tasks IDs.

Lastly there is some good visualization and discussion of what editing does.

Overall this is a good paper with that deals with an interesting problem and proposes an interesting method of increasing performance.

*  I think some ways to strengthen this paper would include addressing some of the comments above, specifically around motivating the "editing" process better and clarifying some of the language around it.
* Finally, the field of continual learning in general could do with moving to more large scale tasks. This would allow testing these methods more thoroughly. While the results here are promising, they do not increase performance uniformly across all tasks -- for example on mini Imagenet which is harder than MNIST -- this method does not help. In general considering larger scale tasks (based on Imagenet, Omniglot or reinforcement learning like Atari, for example) would provide more evidence and allow further improvement of the method.

In the current form however, this is still a very good submission and I would recommend acceptance.

---

> ### Author Response · Authors · 2020-11-20
> **Response to Reviewer #4**
>
> Thanks for your positive comments!
>
> **Q1: One comment I have here is that the way this motivation is presented in the paper can be improved; perhaps with a more clearly written paragraph in the preliminaries. On my first reading of the introduction for example, this sentence for example**
>
> Thanks for pointing out this clarity issue. We will update the updated version of the paper.
>
> **Q2: the reasoning here is that because naturally forgettable examples help during replay, making an example "forgettable" will naturally improve its usefulness. I can't fault this reasoning based on the results but to me it is not naturally clear that this follows. Would the authors be able to provide some intuition for what that means from an optimization perspective?**
>
> Thanks for the question. Given the empirically validated hypothesis of “replaying forgettable examples are more helpful”, we just take the most effective way that aligns with this hypothesis -- i.e., following the gradient of forgetting -- to search in the input space for most forgettable examples. It justifies our editing algorithm, which performs iterative optimization towards finding such forgettable examples.
>
> **Q3: Including results of EWC and MbPA**
>
> Thanks for pointing out! We tried EWC but it does not perform well in a class-incremental learning setup (i.e. a set of class labels appear incrementally in the stream, which is the setting applied for split MNIST, split CIFAR, and split mini ImageNet in our paper.) We will update the results of MbPA when it is ready.
>
>
>
> Finally, we would like provide clarifications regarding the comment "Further, the authors propose an online metric for how much the network forgets an example (by comparing the loss at consecutive timesteps)". We would like to clarify that the technique of estimating forgetting online is applied in a previous work ER-MIR [1] which is discussed and included as one of our baselines; however, there are differences on what the algorithm does after estimating forgetting online: in ER-MIR, the estimated forgetting is used for retrieving most forgettable example from the memory; while our proposed method tries to editing memory example to that they are more forgettable. We compared two methods in the paper and also showed GMED can be seamlessly combined with MIR (GMED) to achieve better performance than MIR because they are mostly orthogonal approaches.
>
> [1] Aljundi, Rahaf, et al. "Online continual learning with maximal interfered retrieval." Advances in Neural Information Processing Systems. 2019.

---

> > ### Comment · AnonReviewer4 · 2020-11-23
> > **Re**
> >
> > Thank you for your responses!

---

### Official Review · AnonReviewer3 · 2020-11-09
**This paper proposes a gradient based method to improve the samples stored in the replay buffer of continual learning algorithms by taking derivatives with respect to the forgetting measure. The paper's idea is nice and intuitive but the presentation and empirical validation can be improved.**

**Rating:** 5
**Confidence:** 5

**Review:**

Differentiation with respect to forgetting looks a promising idea that can complement differentiation with respect to current task accuracy and leads the CL algorithm towards a more stable solution by a good tradeoff between stability and plasticity.

One major concern about this paper is the empirical validations. Some baselines like AGEM are underperforming compared to reported numbers on other papers. The idea of memory editing can also be added as an additional loss to other methods too (not only to ER and MIR). Is that true?

Random edits look quite promising. It may suggest that most of improvement is coming from replay (it can be anything) and the regularization effect that it provides to not to overfit to the data.

Isn't figure 4 telling something trivial? Because each class has a unique pattern of active inputs and that leads similar activation patterns on data and subsequently to a similar gradient profile within the class.

In equation 6 when we replace the forgetting measure (d_t) with equation 5 loss function \ell is going to have a different coefficient (-\alpha-\beta) which basically is a special way of weighing the two loss terms on \theta and \theta'. It's good to elaborate on this more.

The connection of eq. 1 and eq. 2 to te final editing formula (stated in eq. 3) is a little loose. Especially starting from eq. 2 approximation using eq. 3 is not the most natural or organic step.

The paper is well written and easy to follow.
page 2 --->... one online model...
page 3 ---> an categorization ...
page 4 ---> also show that ...
Also note that citations are not properly using parentheses.

---

> ### Author Response · Authors · 2020-11-20
> **Response to Reviewer #3**
>
> Thanks for your insightful comments!
>
> **Q1: Some reported results are underperforming reported results in other papers**
>
> The performance of a method depends on several factors, such as the size of the replay memory, the number of tasks for manually created datasets (e.g. permuted MNIST). We follow experimental setups from [1] for split MNIST, permuted MNIST, split CIFAR-10, and split mini-ImageNet; we follow [2] for rotated MNIST experiments. It is possible that other papers do not use identical settings, which makes it hard to directly compare the results.
>
> We also note that for the MIR baseline, we modified the strategy of sample examples from their memory so that MIR can be run in an online task-free setup (which is discussed in Appendix A: the original MIR filter out memory examples that belong to the same task as the current data stream, which assumes knowledge about tasks boundaries.) We notice that it causes some performance drop on MIR.
>
> **Q2: Can the idea of memory editing also be added as an additional loss to other methods too (not only to ER and MIR)**
>
> Yes, it is true, but probably it requires some modifications. For example, when applying GMED on MIR, we performed adaptation that the algorithm draws a separate set of examples for replay and edit, considering that examples replayed by MIR are already the most forgettable ones.
>
> **Q3: Random editing seems promising. Where does improvement of GMED comes from**
>
> It is certain that replaying examples itself (i.e., the ER baseline) greatly alleviates forgetting compared to simple online training. Our method builds upon ER and tries to improve the performance of it. The minor improvement of random edit over ER or MIR demonstrates that regularization effects (or data augmentation) can be a reason behind the performance improvement of GMED. However, we note that editing examples following GMED outperform over alternative data augmentation baselines. We include the results in our response to Reviewer #2.
>
> **Q4: The regularization is a special way of weighing the two loss term**
>
> Thanks for pointing out! We included this in the updated version of the paper.
>
> **Q5: formatting issues and typos**
>
> Thanks for pointing out! We will update them in the updated version.
>
> [1] Aljundi, Rahaf, et al. "Online continual learning with maximal interfered retrieval." Advances in Neural Information Processing Systems. 2019.
> [2] Chaudhry, Arslan, et al. "Using hindsight to anchor past knowledge in continual learning." arXiv preprint arXiv:2002.08165

---

### Decision · Program_Chairs · 2021-01-07
**Final Decision**

**Decision:**

Reject

**Comment:**

The range of the initial reviews was fairly high with overall scores ranging from 4 to 7.

The authors provided a good response that answered most of the reviewers' comments and questions. One of the reviewers even increased their score following the authors' response.

The focus of some of our discussions and what ultimately led to my suggestion was the related work of MIR [1]. The methodological differences between (the three versions of) MIR in [1] and GMED [this paper] appear less significant than what the current submission suggests. While there is some disagreement between the authors and Reviewer1 about the exact differences, I find that the current manuscript does not acknowledge the close relationship between these two contributions. Further, from the experimental standpoint and without further justifications the gains from GMED+MIR could be attributed to using more replay (from combining GMED and MIR).


In their response, the authors disputed the view of Reviewer1. I believe the source of the confusion between the author and the reviewer might be captured in this sentence from the author response to Reviewer1: The approach does not learn a generator that can “generate examples that are more forgettable for the classifier”; instead, feeding more forgettable examples in GEN-MIR aims at reducing the forgetting of the generator.

Looking at Equations 2, 3 and Algorithm 1 from [1], in GEN-MIR while two different procedures are used to obtain forgettable examples for the generator (B_G in Alg. 1) and the classifier (B_C in Alg. 1), the generator is used in both cases. In other words, the generator is used to generate examples for both itself and for the classifier. So, I think it's fair to say that the generator does indeed generate examples that are more forgettable for the classifier (Eq. 2).


I strongly encourage the authors to prepare another version of their work where the differences between MIR [1] and their contribution are clearly highlighted and the results show the advantages of GMED (including memory-editing in data space).


[1] Rahaf Aljundi, Lucas Caccia, Eugene Belilovsky, Massimo Caccia, Min Lin, Laurent Charlin, and Tinne Tuytelaars. Online continual learning with maximally interfered retrieval. In NeurIPS 2019. https://arxiv.org/abs/1908.04742